# The Work and Social Adjustment Scale (WSAS): An investigation of reliability, validity, and associations with clinical characteristics in psychiatric outpatients

Jakob Lundqvist[1]*, Martin Schevik Lindberg[1,2], Martin Brattmyr[1], Audun Havnen[1,3], Odin Hjemdal[1], Stian Solem[1]

1 Department of Psychology, Norwegian University of Science and Technology (NTNU), Trondheim, Norway, 2 Mental Healthcare Services, Trondheim Municipality, Trondheim, Norway, 3 Division of Psychiatry, Nidaros Community Mental Health Centre, St. Olavs University Hospital, Trondheim, Norway

* jakob.lundqvist@ntnu.no

**Data Availability Statement:** All WSAS data have been de-identified and openly uploaded to OpenICPSR (https://doi.org/10.3886/E198102V1).

## Abstract

### Objective

This study, the first to assess the reliability and validity of the Work and Social Assessment Scale (WSAS) in Norwegian routine mental health care, examines differences in functional impairment based on sick leave status, psychiatric diagnosis, and sex.

### Method

Including 3573 individuals from community mental health services ($n_1$ = 1157) and a psychiatric outpatient clinic ($n_2$ = 2416), exploratory factor analysis (EFA) on subsample 1 and confirmatory factor analysis (CFA) on subsample 2 were utilized to replicate the identified factor structure.

### Results

EFA supported a one-factor model, replicated by the CFA, with high internal consistency ($\alpha$ = .82, $\omega$ = .81). Patients on sick leave reported greater impairments in all aspects of functioning, except for relationships, with the largest effect size observed in the reported ability to work ($d$ = .39). Psychiatric outpatients with major depressive disorder were associated with difficulties in home management, private leisure activities, and forming close relationships. Patients with attention-deficit/hyperactivity disorder reported less impairment than those with other disorders. Patients with personality disorders reported more relationship difficulties than those with PTSD, ADHD, and anxiety. No differences were found in the perceived ability to work between diagnoses. Women had a higher impairment in private leisure activities, whereas men reported more impairment in relationships.

The deposited data exclude demographic information. We are constrained from sharing demographic details as it involves potentially identifiable and sensitive patient information. Moreover, the patients have not provided consent for the open sharing of such data.

**Funding:** The funder provided support in the form of salaries for the first author (JL) but did not have any additional role in the study design, data collection, and analysis, decision to publish, or preparation of the manuscript. The specific roles of the authors are articulated in the 'author contributions' section.

**Competing interests:** The authors have declared that no competing interests exist.

## Conclusion

The demonstrated reliability and validity suggest that WSAS is a valuable assessment tool in Norwegian routine mental health care. Variations in functional impairment across sick leave status, sex, and psychiatric diagnoses highlight the importance of integrating routine assessments of functional impairment into mental health care practices. Future research should combine WSAS with register data to allow for a broader understanding of treatment effectiveness, emphasizing improvements in functional outcomes alongside symptom alleviation.

## Introduction

Functional impairment in patients with mental disorders has received increased attention and has been argued to be a more important treatment outcome than symptoms [1]. Mental disorders often affect an individual's physical, mental, and social aspects and are associated with substantial functional impairment, such as impaired work ability and social functioning [2,3]. This often leads to absenteeism, long-term disability claims, and financial burdens [4–6].

In Norway, mental disorders represent 26% of all certified sick leave days [7], but the proportion of individuals suffering from mental disorders is likely higher since mental disorders are often under-reported as a reason for sick leave [8]. Moreover, mental disorders are frequently recurrent and carries a high risk of relapse after treatment [9]. Sick leave due to mental disorders tends to be longer in duration than sick leaves in general [10]. Being out of work is associated with a less favorable treatment prognosis [11]. Still, in mental healthcare, there is a tendency to prioritize symptom reduction over functional impairments, which are frequently treated as a secondary concern [12].

Functional impairment may vary across mental disorders. Individuals diagnosed with personality disorders like borderline personality disorder have reported significantly higher levels of functional impairment than individuals with depression [13]. Differences in impairment between disorders could be due to both physical and psychological comorbidity, severity, and chronicity, but it is also important to note that the use of different instruments for measuring impairment could affect the results [14,15]. Adults with attention-deficit/hyperactivity disorder (ADHD) have reported greater functional impairments in social relationships, daily life functioning, and academic performance than those with other mental disorders [16]. While no differences were observed in the duration of unemployment, patients with ADHD were more frequently on sick leave and less likely to be unemployed than other patients.

Furthermore, previous studies exploring sex differences in functional impairment have yielded conflicting outcomes. While some did not find any significant differences between sexes among patients with personality disorders [17] and chronic fatigue syndrome [18], others found that male patients with personality disorders in a Norwegian sample reported less social leisure impairment but faced greater challenges in establishing and maintaining close relationships compared to women [19].

Although functional impairment may be improved through work-focused interventions, symptoms of mental disorders do not necessarily improve, and similarly, improved symptoms do not necessarily directly result in enhanced functional ability [20–22]. Hence, a delay in functional improvement compared to symptomatic improvement has been noted [23–25]. Improvements in work ability and social adjustment are important indicators of employment

status and predictors of long-term remission [26,27]. Diagnostic criteria for mental disorders imply functional impairment, yet cross-evaluation of functional impairment is infrequently performed [28]. The interplay between symptoms and functioning can be characterized as a combined parallel and serial process [29]; therefore, the assessment of work and social functioning should supplement standard symptom measures [30].

The Work and Social Adjustment Scale (WSAS) was developed as a brief measure of occupational and social functioning. Previous studies have found robust psychometric properties [19,31], with unidimensionality, sensitive to change, and good test-retest stability [31,32]. Accordingly, WSAS has been included in clinical guidelines and protocols as a valid tool to assess the level of disability or impairment in daily life due to mental or physical health conditions [33]. The utility of WSAS has been demonstrated in a range of psychiatric and somatic disorders in various settings [31,34]. Since it is short, easy to comprehend, and quick to complete, WSAS is recommended for both assessment and treatment evaluation [18,35].

Previous studies of WSAS have predominantly focused on mental health conditions such as depression [31,36], anxiety disorders [37], mixed depression and anxiety disorders [37,38], and social phobia [34] in specialized clinics, limiting the generalizability of findings to routine clinical settings. In routine clinical practice, patient populations are often heterogeneous, reflecting the complexity of mental health conditions encountered in real-world scenarios. The significance of our study lies in its exploration of WSAS in a routine clinical context, being the first to shed light on the reliability and validity of this instrument assessing functional impairment in heterogeneous samples in Norwegian routine mental health care [39]. Furthermore, there is a lack of prior research investigating the association between WSAS items and sick leave.

To our knowledge, only one study has previously incorporated sick leave data, to study the capacity of WSAS to differentiate between individuals on sick leave and those actively employed [40]. The study, which included a cohort of Danish patients with emotional disorders ($N = 230$), reported that the total WSAS score displayed low specificity (74%) and sensitivity (55%) in predicting long-term sick leave, with the optimal cut-off point identified at 23.

By encompassing two large and heterogeneous samples from the Norwegian public routine mental health care system, spanning both community mental health services and psychiatric outpatient care, this study seeks to explore the reliability and validity of WSAS in routine clinical practice. Additionally, we intend to investigate the association with sex, sick leave status, and psychiatric diagnoses. With an expectation of robust psychometric properties for WSAS within the Norwegian routine mental health care setting, our objective is to explore its correlation with sick leave status. This evaluation will not only contribute to the refinement of the instrument but also hold the potential to inform evidence-based interventions, ultimately aiming to improve functional outcomes in routine mental health care settings.

## Method

### Participants and procedure

The data used for this study were obtained from a quality assessment project focusing on routine care provided to treatment-seeking adults who were either referred to a psychiatric outpatient clinic by general practitioners or sought help within the community mental health services. All responses were recorded at start of treatment. Data from the psychiatric outpatient clinic were collected between February 2020 and February 2022, and data from the community mental health service were collected between September 2020 and October 2022. There were no specific exclusion criteria, but patients receiving treatment in special units (e.g., patients with obsessive-compulsive disorder, schizophrenia, substance abuse, and retired

patients) did not take part in the study. The respondents provided their informed consent to participate and submitted their responses using a web-based portal (checkware.no). This study was approved by the Regional Committee for Medical and Health Research Ethics (REK; reference number 2019/31836) and the Norwegian Centre for Research Data (NSD; reference number 2020/605327).

The total sample consisted of 3573 outpatients with a mean age of 31.9 years ($SD = 11.27$) and the majority were women ($n = 2312$; 65%). Retired persons ($n = 6$) were excluded from this study. The total sample consisted of two subsamples. Subsample 1 contained individuals seeking help at the community mental health service ($n_1 = 1157$). This subsample included patients from a low-threshold service ($n = 866$) and a referral-based service ($n = 291$).

Subsample 2 included individuals referred to a psychiatric outpatient clinic ($n_2 = 2416$). In this subsample, 1476 patients (60%) were diagnosed according to ICD-10. The main diagnosis was grouped according to the ICD-10 structure and the six most common diagnoses within the subsample were selected ($n = 1051$). Those who had received several main diagnoses during the treatment period ($n = 203$) were categorized as comorbid. The patients with one main diagnosis had depression (F32.0-F33.9, $n = 249$, 29.4%), ADHD (F90.0-F90.8, $n = 221$, 26.1%), anxiety disorders (F40.0-F43.9, $n = 130$, 15.3%), PTSD (F43.0-F43.9, $n = 107$, 12.6%), personality disorders (F60.0-F60.9, $n = 90$, 10.6%), or bipolar disorder (F31.0-F31.9, $n = 51$, 6%). For further descriptive information, see Table 4.

## Setting

The participants received routine care within the Norwegian public mental health care system. Subsample 1, the community mental health service, offered a low-threshold service for mild to moderate mental health issues, focusing on early intervention and support. In addition, the community service offered referral-based services for individuals with moderate mental health problems and complex life challenges, who may have had previous contact with psychiatric outpatient services. The community service offers counseling, therapy, and support, all aimed at promoting mental well-being and addressing mental health concerns at an earlier stage. Subsample 2 was referred to psychiatric outpatient care, which focuses on moderate to severe mental health problems, providing specialized assessment, diagnosis, and treatment. The two levels of care aim to ensure that individuals receive appropriate care based on the severity and complexity of their mental health concerns, improving overall well-being, and promoting mental health across the population.

## Measures

WSAS is a short and generic self-report designed to assess the patient's functional impairment related to their mental disorder [31]. The five-item scale addresses impairments in work/ study, home, social life, private leisure activities, and interpersonal relations. Each item is self-rated on a nine-point Likert scale that ranges from "Not at all" (0) to "Severely impaired" (8) giving a total score ranging from 0 to 40. Scores above 20 indicate severe functional impairment, scores from 10–20 moderate impairment, while scores below 10 are viewed as subclinical [31]. Previously, the Global Assessment of Functioning (GAF) scale was utilized in this psychiatric outpatient clinic; however, it was discontinued due to its unreliability in routine clinical settings [41]. WSAS was selected for clinical use based on its recommendation in clinical guidelines as a dependable instrument for evaluating disability or impairment arising from mental or physical health conditions [33]. The brevity, simplicity, and rapid completion time of WSAS ($M = 1.5$ min., $SD = 1.3$) have made it a recommended choice for both assessment and treatment evaluation [18,35].

Information on sex, age, and work status was collected. Work status was self-reported and converted from free text to binary values of either currently in work or on 100% sick leave. The sick leave category also included people on work assessment allowance, a social security benefit providing financial support, and vocational rehabilitation. Unpublished data from the two clinics suggest that the most common reasons for being on sick leave at the start of treatment were psychological problems (79.1%), musculoskeletal disorders (5.3%), neurological disorders (3%), and general and unspecified conditions (3%) like fatigue, fever, or general weakness.

## Statistical analyses

This study employed an exploratory cross-sectional design to assess the reliability and validity of WSAS across all its items in Norwegian routine mental health care, while also examining variations in functional impairment based on sick leave status, psychiatric diagnosis, and sex.

In the total sample, 96.1% answered all five WSAS items. Out of the missing (3.9%), 112 participants did not answer a single WSAS item, and these respondents were removed from further analysis. The analysis of missing data patterns indicated that missing was completely at random (MCAR), as confirmed by Little's test ($p > 0.05$).

Since studies of WSAS factor structure with heterogeneous patients from routine care facilities are scarce, we utilized exploratory factor analysis (EFA) in subsample 1, patients in community services, and confirmatory factor analysis (CFA) in subsample 2, a set of patients treated in a psychiatric outpatient clinic. The EFA was executed with an orthogonal varimax rotation to determine its suitability. The factor structure identified in subsample 1 was replicated with CFA for subsample 2. This dual-method approach serves both to explore (EFA) and to confirm (CFA) the factor structure in two subsamples across diverse routine mental health care settings. It is a common procedure if a measure is tested in a new setting [42]. To further validate the robustness of the findings, supplementary analyses were conducted. Subsamples $n_1$ and $n_2$ were combined and then randomly split into two (to represent the full spectrum of mental health issues). An EFA was then performed using half the sample ($n_3$) and CFA on the other ($n_4$).

Model fit was evaluated using the Root Mean Square Error of Approximation (RMSEA), the Standard Root Mean Residual (SRMR) fit indices, and the comparative fit indices Comparative Fit Index (CFI) and Tucker-Lewis Index (TLI). RMSEA, sensitive to model misspecification, measures the disparity between observed data and the model. SRMR measures the average standardized difference between observed and predicted correlations, while CFI and TLI assess the model's replication of the observed covariance structure [42].

RMSEA values less than .06 typically indicate a good model fit [43]. Similarly, SRMR values less than .05 indicate good model-data fit [44]. Thresholds of .95 or greater for CFI and TLI were considered indicative of a good fit [43]. Values less than .08 suggest a satisfying model fit for SRMR [45]. To examine whether a one-factor structure or a two-factor structure best fitted the data, two models were evaluated: the original one-factor model (Model 1) and a second model (Model 2) that incorporated a post-hoc adjustment to permit a correlation between the error terms of items 3 and 5, as suggested by the modification indices. Composite reliability, which has been suggested as a more robust measure than others [46], was utilized to assess internal consistency. A range of .7 to .9 was considered acceptable for satisfactory internal consistency. Further, differences were examined between groups (sick leave status, diagnoses, and sex) using one-way ANOVA with post-hoc analyses and t-tests. Mplus model option command was used for examining measurement invariance across gender, sick leave, and age [47]. Configural invariance was confirmed if the number of factors and indicator-factor patterns

were equivalent across groups. Metric invariance was established by constraining factor loadings to be equal across groups, while scalar invariance involved constraining both factor loadings and thresholds to be equal. Additionally, models were scrutinized based on changes ($\Delta$) in fit indices, with $\Delta$CFI $\geq$ -0.01 and $\Delta$RMSEA $<$ 0.015 as recommended thresholds [48]. The Maximum Likelihood (ML) estimator was used for both EFA and CFA.

# Results

## Validity

The Kaiser-Meyer-Olkin (KMO) test showed an overall KMO of .83, indicating an adequate sampling in Subsample 1. All subjects were included in this analysis. The results of an orthogonal varimax rotation of the solution indicated a one-factor solution with an eigenvalue of 2.42. When rotated, Factor 1 explained 48% of the variance of the total sample with factor loadings from .67 to .71. The results from comparing scree plots and eigenvalues (i.e., eigenvalues $>$ 1.00) suggested that the one-factor solution was a better fit for the data than the two- and three-factor solutions. This conclusion was also supported by the results of the parallel analysis. The eigenvalues of the remaining factors in the factor solution were below one, indicating that they contributed very little to the explained variance. See Table 1 for factor loadings across items, sex, and forms of community services.

Using EFA when combining subsamples 1 and 2 revealed near identical results (see S1 Table). The KMO test was .81 for sample $n_3$, and an orthogonal varimax rotation indicated a 1-factor solution with an eigenvalue of 2.39. When rotated, Factor 1 explained 48% of the variance of the total sample with factor loadings ranging from .64 to .74. The results from comparing scree plots and eigenvalues (i.e., eigenvalues $>$ 1.00) suggested that the one-factor solution was a better fit for the data than the two- and three-factor solutions.

CFA was used for modeling the unifactory solution in subsample 2. The RMSEA value indicated a poor model fit to the data (RMSEA = .14) and the SRMR value suggested an indicative of a close-fitting model (SRMR = .05), resulting in unsatisfactory fit statistics. Correlated residuals were observed for Items 3 and 5 in Model 1 $\chi^2$ (5, N = 2416) = [253.54], ($p < .001$), (90% CI: 0.13–0.16), The Comparative Fit Index (CFI = .94), Tucker-Lewis Index (TLI = .87). The modification indices suggested a residual item correlation between items 3 and 5 which was added to Model 2. This adjustment aligns with the theoretical perspective, as items 3 and 5 specifically inquire about the impairment of social relatedness. The Model 2 yielded an acceptable model fit to the data (RMSEA = .05) with a (Stdyx total $\delta$ = .35, [$p < .001$]. $\chi^2$(4, 2416) = 32.10, $p < .001$, (90% CI: 0.04–0.07), CFI = .99, TLI = .98. In the single-factor WSAS model, all factor loadings were positive and substantial ($p < .001$), with values ranging between r = .57 and .77. These results indicate that the factor loadings were adequate for all items and Model 2 was selected as the final model.

**Table 1. Factor loadings for WSAS by group and sex in subsample 1 ($n_1$).**

| WSAS Item. Impaired: | Community: Total subsample ($n_1$ = 1157) | Community: Low-threshold service ($n$ = 601) | Community: Referred ($n$ = 187) | Men ($n$ = 252) | Women ($n$ = 788) |
|---|---|---|---|---|---|
| 1. Ability to work | 0.64 | 0.57 | 0.66 | 0.63 | 0.64 |
| 2. Home management | 0.71 | 0.69 | 0.71 | 0.70 | 0.72 |
| 3. Social leisure activities | 0.71 | 0.72 | 0.75 | 0.71 | 0.70 |
| 4. Private leisure activities | 0.67 | 0.68 | 0.62 | 0.66 | 0.68 |
| 5. Close relationships | 0.66 | 0.64 | 0.67 | 0.70 | 0.64 |

*Note.* WSAS = The Work and Social Adjustment Scale.

CFA was also used for modeling the unifactory solution (model 1) in subsample 4 ($n_4$ = 1787). The results are summarized in S2 Table and were nearly identical to the original analysis. The RMSEA value indicated a poor model fit to the data (RMSEA = .12) and the SRMR value suggested an indicative of a close-fitting model (SRMR = .04), resulting in unsatisfactory fit statistics in $n_4$. As with the original analysis, correlated residuals were observed for Items 3 and 5 (social relatedness) which were added to Model 2. Model 2 yielded an acceptable model fit to the data $x^2$(4, 1787) = 18.49, $p < .001$, RMSEA = .05, CFI = .99, TLI = .99, SRMR = .05. All factor loadings were positive and substantial ($p < .001$), with values ranging between $r$ = .36 and .56.

The model was fitted separately across sex, age, and sick leave in the test of measurement invariance (see Table 2). High and low age was categorized as above or below the median age (27 years). Fit indices indicated a good model fit. Configural invariance was established as the one-factor structure exhibited satisfactory model fit in all subgroups. The metric model, with factor loadings constrained to be equal across women and men, high and low age, and those on sick leave and those in work, demonstrated no deterioration in fit indices and was retained. In the last step, factor loadings and item thresholds were constrained to be equal across women and men, age groups, and sick leave status to assess scalar invariance. However, this model exhibited a minor decline in fit indices, indicating partial invariance and highlighting limitations in comparing latent means. These results suggest that these different groups may perceive WSAS items differently, emphasizing the importance of studying WSAS at the item level.

## Reliability

The internal consistency among the items was good (Cronbach's $\alpha$ = .82, 95% CI .80 to .84) in subsample 1 with a mean inter-item correlation of .48 and in subsample 2 (composite reliability $\omega$ = .81) with a mean inter-item correlation of .47. Additionally, all items exhibited a good adjusted item-scale correlation ($r$ values >.40).

## Group comparisons

Group comparisons showed significant differences between patients on sick leave compared to patients at work. This applied to items 1–4, and for the total score (see Table 3). Patients on sick leave reported more impairment in all aspects of functioning except for close

**Table 2. Measurement invariance of WSAS across sex, age, and sick leave status.**

| | $\chi^2$ (df) | CFI | RMSEA [90% CI] | $\Delta\chi^2$ (df) | p | ΔCFI | ΔRMSEA |
|---|---|---|---|---|---|---|---|
| *Sex* | | | | | | | |
| Configural | 33.687 (8) | .993 | .052 [.034 –.070] | — | — | — | — |
| Metric | 34.436 (12) | .994 | .039 [.024 –.055] | 0.748 (4) | < .001 | .001 | -.013 |
| Scalar | 76.507 (16) | .984 | .056 [.044 –.069] | 42.071 (4) | < .001 | -.010 | .017 |
| *Age* | | | | | | | |
| Configural | 37.460 (8) | .992 | .055 [.038 –.074] | — | — | — | — |
| Metric | 41.196 (12) | .992 | .045 [.030 –.060] | 3.736 (4) | .443 | 0 | -.010 |
| Scalar | 128.258 (16) | .971 | .076 [.064 –.089] | 87.062 (4) | < .001 | -.021 | .031 |
| *Sick leave* | | | | | | | |
| Configural | 36.514 (8) | .992 | .057 [.039 –.076] | — | — | — | — |
| Metric | 42.549 (12) | .991 | .048 [.033 –.064] | 6.035 (4) | .197 | -.001 | -.009 |
| Scalar | 131.216 (16) | .966 | .081 [.068 –.094] | 88.667 (4) | < .001 | -.025 | .033 |

**Table 3. Group comparisons on WSAS items ($n_2$).**

| Item | On 100% sick leave (A) (n = 661) M (SD) | In work (B) (n = 1561) M (SD) | d | t | Comp. | p |
|---|---|---|---|---|---|---|
| WSAS Total score | 23.14 (7.98) | 20.47 (8.29) | .30 | 7.03 | A>B | < .001 |
| 1. Ability to work | 5.51 (1.92) | 4.64 (2.08) | .39 | 9.18 | A>B | < .001 |
| 2. Home management | 4.47 (2.07) | 3.79 (2.13) | .15 | 3.58 | A>B | < .001 |
| 3. Social leisure activities | 5.06 (2.07) | 4.31 (2.20) | .33 | 7.77 | A>B | < .001 |
| 4. Private leisure activities | 4.58 (2.21) | 3.64 (2.33) | .22 | 5.08 | A>B | < .001 |
| 5. Close relationships | 4.78 (2.26) | 4.09 (2.27) | .05 | 1.29 | NS | .196 |
| Item | Men (M) (n = 892) M (SD) | Women (W) (n = 524) M (SD) | d | t | Comp. | p |
| WSAS Total score | 21.09 (8.52) | 21.47 (8.19) | .04 | - 1.08 | NS | .278 |
| 1. Ability to work | 4.92 (2.15) | 4.93 (2.04) | .01 | - 0.16 | NS | .874 |
| 2. Home management | 3.85 (2.16) | 3.93 (2.09) | .03 | - 0.85 | NS | .396 |
| 3. Social leisure activities | 4.48 (2.38) | 4.59 (2.12) | .05 | - 1.13 | NS | .261 |
| 4. Private leisure activities | 3.53 (2.31) | 3.96 (2.29) | .18 | - 4.35 | M<W | < .001 |
| 5. Close relationships | 4.30 (2.36) | 4.06 (2.22) | .10 | 2.46 | M>W | .014 |

*Note.* WSAS = The Work and Social Adjustment Scale. *d* = Cohen's d. Comp = Comparison across groups, NS = Not significant.

relationships. The largest effect size was observed in the reported ability to work (*d* = .39). There were also sex differences as women reported more impairments in private leisure activities, while men had more impairments regarding maintaining close relationships.

In Subsample 2, differences in WSAS scores across diagnoses were found (see Tables 4 and 5). As noted, 72% of patients with depression reported severe functional impairment compared to 44% of those with ADHD. In sum, the post hoc analysis indicated that patients with depression reported more difficulties with home management and the ability to do things alone, as well as forming and maintaining close relationships, compared to patients with anxiety disorders and ADHD. No differences were found between those with several psychiatric diagnoses and those included in one specific diagnostic group, except for those with ADHD. Patients with ADHD reported lower impairments compared to all other patient groups. Patients with personality disorders reported more difficulties with forming and maintaining close relationships compared to patients with PTSD, ADHD, and anxiety. There were no differences between diagnosis groups in their self-rated ability to work.

As shown in Table 4, a majority of patients receiving routine mental health care exhibited severe functional impairment, irrespective of their psychiatric diagnosis, sick leave status, or sex. These findings align with expectations, given that individuals seeking treatment often experience higher levels of impairment and symptoms than the general population.

## Discussion

This study aimed to explore the reliability and validity of WSAS and investigate the association with sick leave status, psychiatric diagnoses, and sex in two large heterogeneous samples of routine mental health care patients. As anticipated, our findings demonstrated good internal consistency of WSAS, supporting its utility in assessing functional impairment across a heterogeneous sample in the initial stages of routine public treatment. Notably, participants on sick leave reported more functional impairment compared to working individuals. Variations in

**Table 4. Levels of functional impairment by level of care, age, diagnosis, and sick leave status.**

| Variables | Mild functional impairment < 10 *n* (%) | Moderately functional impairment 10–20 *n* (%) | Severe functional impairment > 20 *n* (%) | Total *n* |
|---|---|---|---|---|
| *Level of care* | | | | |
| Community services ($n_1$) | 163 (14.1) | 457 (39.5) | 537 (46.4) | 1157 |
| Low-threshold service | 142 (16.4) | 383 (44.2) | 341 (39.4) | 866 |
| Referred | 21 (7.2) | 74 (25.4) | 196 (67.3) | 291 |
| Psychiatric outpatient clinic ($n_2$) | 235(9.7) | 804 (33.3) | 1377 (56.9) | 2416 |
| *Age group (N)* | | | | |
| 18–24 | 107 (9.7) | 409 (37.1) | 585 (53.1) | 1101 |
| 25–29 | 92 (10.9) | 296 (35.2) | 451 (53.7) | 839 |
| 30–34 | 53 (9.8) | 205 (38.1) | 272 (50.6) | 537 |
| 35–39 | 40 (12.4) | 117 (36.3) | 165 (51.2) | 322 |
| 40–49 | 49 (11.8) | 126 (30.3) | 240 (57.8) | 415 |
| 50+ | 53 (16.0) | 96 (29.0) | 182 (54.9) | 331 |
| *Sex ($n_2$)* | | | | |
| Men | 95 (10.7) | 290 (32.5) | 507 (56.8) | 892 |
| Women | 140 (9.2) | 514 (33.7) | 870 (57.1) | 1524 |
| *Sick leave status ($n_2$)* | | | | |
| On 100% sick leave | 44 (6.7) | 177 (26.8) | 440 (66.6) | 661 |
| In work | 169 (10.8) | 572 (36.6) | 820 (52.5) | 1561 |
| *Diagnoses ($n_2$)* | | | | |
| Depression | 13 (4.1) | 77 (24.2) | 228 (71.7) | 318 |
| ADHD | 30 (13.6) | 93 (42.1) | 98 (44.3) | 221 |
| Anxiety | 10 (8.5) | 75 (39.9) | 103 (54.8) | 188 |
| PTSD | 10 (8.5) | 36 (30.5) | 72 (61.0) | 118 |
| Personality disorders | 5 (4.6) | 28 (25.7) | 76 (69.7) | 109 |
| Bipolar disorder | 5 (7.5) | 18 (26.9) | 44 (65.7) | 67 |
| Comorbid | 8 (4.4) | 52 (28.7) | 121 (66.9) | 181 |

*Note*. ADHD = attention-deficit/hyperactivity disorder, PTSD = Post-traumatic stress disorder.

impairment were also observed across different psychiatric diagnoses, where individuals with depression reported the highest levels, while those with ADHD reported the lowest. Furthermore, sex disparities occurred, as women reported greater impairments in private leisure activities, while men encountered more challenges in maintaining close relationships.

WSAS is a widely used measure of functional impairment, but its reliability and validity have not previously been explored in a large heterogeneous sample across different levels of routine mental health care. The study found a unidimensional structure of WSAS (.82 and .81), consistent with previous research on psychiatric samples (.79 and .89) [17,16] and the factor structure reported by the originators [31]. However, most of the previous studies investigated the factor structure of WSAS in specific patient populations, whereas two heterogeneous outpatient samples were included in the current study. The one-factor model here identified using EFA in a community sample was replicated using CFA in a psychiatric outpatient sample. The observed RMSEA value, sensitive to model complexity, suggested a poor fit for the unadjusted model, a discrepancy that can be traced back to error variance between items 3 and 5, influencing the RMSEA. However, when the residuals were allowed to correlate in an adjusted model, this resulted in an acceptable model fit. This adjustment also is theoretically

**Table 5. Means and standard deviations of WSAS scores across diagnoses ($n_2$ = 1051).**

| Item | F | p | Depression (n = 249) | Personality disorders (n = 90) | Bipolar disorder (n = 51) | PTSD (n = 107) | Anxiety disorders (n = 130) | ADHD (n = 221) | Comorbid (n = 203) | Comp. |
|---|---|---|---|---|---|---|---|---|---|---|
| | | | (A) | (B) | (C) | (D) | (E) | (F) | (G) | |
| WSAS Total score | 10.7 | < .001 | 24.26 (7.25) | 24.08 (7.90) | 22.51 (8.51) | 22.02 (7.69) | 20.89 (7.34) | 19.23 (8.42) | 23.34 (7.32) | A>EF F<ABDG |
| 1. Ability to work | 2.1 | .05 | 5.37 (1.94) | 5.24 (2.09) | 4.90 (2.25) | 5.23 (1.98) | 4.87 (2.00) | 4.84 (2.03) | 5.24 (1.86) | NS |
| 2. Home management | 4.6 | < .001 | 4.47 (1.99) | 4.19 (2.17) | 4.61 (1.83) | 3.67 (1.96) | 3.60 (1.91) | 3.94 (2.14) | 4.29 (2.05) | A>DE |
| 3. Social leisure activities | 10.7 | < .001 | 5.06 (1.92) | 5.10 (2.20) | 4.78 (2.23) | 5.06 (1.93) | 4.98 (2.03) | 3.74 (2.35) | 4.86 (2.00) | F<ABCDEG |
| 4. Private leisure activities | 10.9 | < .001 | 4.58 (2.09) | 4.34 (2.15) | 3.67 (2.30) | 4.00 (2.18) | 3.42 (2.03) | 3.17 (2.39) | 4.34 (2.30) | A>EF F<ABDG G>E B>E |
| 5. Close relationships | 11.5 | < .001 | 4.78 (2.06) | 5.20 (2.13) | 4.55 (2.21) | 4.06 (2.33) | 4.05 (2.39) | 3.54 (2.17) | 4.61 (2.11) | A>EF F<ABG B>DEF |

*Note.* WSAS = The Work and Social Adjustment Scale. ADHD = attention-deficit/hyperactivity disorder, PTSD = Post-traumatic stress disorder. Comp. = Comparison across psychiatric diagnosis, NS = Not significant.

sound as both items address the impairment of social relatedness. Additionally, we found support for configural and metric invariance across age, sick leave, and sex, however, the results for scalar invariance were unclear. This latter finding corroborates previous investigations of measurement invariance for WSAS [19], thus future studies should investigate measurement invariance across groups of respondents. This study offers strong support for the construct validity of WSAS as a measure of work and social impairment in diverse patient samples with mental health problems and extends the current body of research. The results also supported the good internal consistency of WSAS. The Cronbach's α for Subsample 1 and the composite reliability score for Subsample 2 were 0.82 and 0.81, respectively, aligning with Nunnally's criteria for interpreting Cronbach's α. These criteria suggest that values between 0.8 and 0.9 should be regarded as indicative of good internal consistency [32]. These findings align with prior studies on outpatients in Denmark [40], Germany [36], Norway [19], and the UK [31], focusing on emotional disorders, personality disorders, depression and OCD, respectively.

Furthermore, as anticipated the study found that patients on sick leave were more likely to experience higher levels of self-reported functional impairment than those in work. This pattern was consistent across all items except for the ability to maintain close relationships. Existing literature highlights women's tendency to place greater importance on interpersonal relationships [49], while men report greater challenges in initiating and sustaining such relationships [38]. Previous studies have observed a correlation between functional impairment in women and difficulties in participating in private leisure activities [19]. Our study aligns with these trends, indicating that impaired ability to engage in private leisure activities could be a more relevant indicator of functional impairment in women than in men. This underscores the importance of assessing both the prevalence of functional impairment and relational struggles among both men and women. However, it is noteworthy that some studies have not observed these sex differences [17,18], warranting further research. The current study extends the previous literature on WSAS by demonstrating an association between sick leave status and functional impairment, which provides support for the construct validity of WSAS in relation to sick leave.

Patients with depression, PTSD, personality disorders, bipolar disorder, and anxiety disorders exhibited a mean total WSAS score above 20 at the start of treatment, indicating moderately to severe functional impairment. These findings align with prior research involving patients with depression [50], PTSD [51], and personality disorders [19], thus supporting the anticipated outcome among patients with personality disorders. This underscores the pivotal role of interpersonal dysfunction and relationship impairment in patients with personality disorders, thereby supporting the validity of WSAS in clinical settings [52]. Furthermore, patients with anxiety disorders reported a mean WSAS score of 21 indicating a moderately to severe level of functional impairment, while previous research has reported a mean score of 13–16, reflecting moderate impairment [51]. The inclusion of a broad range of anxiety disorders in the current study, extending beyond specific diagnoses may have contributed to the relatively higher severity level found when compared to Silove et al. [53], where certain anxiety disorders were referred to other specialist clinics in the latter study.

In line with a previous study [16], there were no differences in self-reported ability to work across the various diagnoses, indicating that regardless of the specific diagnosis, mental disorders negatively impact one's capacity to work. Previous research has underlined the positive aspects of work for mental health [54,55]. As such, WSAS may be a useful screening tool for identifying an individual's perception of their ability to work and may be useful as an indicator when treating patients with mental disorders.

Previous findings indicate that patients with ADHD have greater functional impairments in social relationships, daily life functioning, and academic performance when compared to individuals with depression, personality disorders, and bipolar disorder [16]. In contrast, the current study indicates that patients with depression had a high impairment, while patients with ADHD exhibited the lowest functional impairment of the mental disorders examined. There may be several explanations for these conflicting results. The aforementioned study [16] had a smaller sample size and measured functional impairment with a diagnostic interview for symptoms severity of ADHD, which may to a lesser extent assess patients' functional impairment [56]. In contrast, the current study included a large sample and included both clinician-assessed diagnoses and patients' self-reported level of functional impairment.

According to Norwegian guidelines, psychiatric outpatient clinics should prioritize individuals with severe ADHD symptoms and significant functional impairment at school, work, and home [57]. Unlike for depression and anxiety disorders, there are no specific guidelines for ADHD patients with mild or moderate impairment. The moderate functional impairment level reported by patients with ADHD in the current study was therefore unexpected. A possible explanation is that clinicians and patients assess impairment differently. Research indicates that an underreporting of symptoms may contribute to variations in reported functional impairment among individuals with ADHD compared to other disorders [56,57]. Another possible explanation is that individuals with a suspected diagnosis of ADHD are more often referred by their GP to psychiatric outpatient clinics for a diagnostic assessment of ADHD. Furthermore, the escalating prevalence of ADHD diagnoses, which has been argued to be influenced by changes in diagnostic criteria and heightened awareness, prompts inquiry into whether alterations in diagnostic criteria could account for observed discrepancies compared to earlier studies [58,59]. Additional research is needed to understand the variability of impairments across mental disorders, as this has clinical relevance both for the allocation of treatment resources and better targeting of patient impairments across subgroups of patients.

It is worth noting that the total sample consisted predominantly of young individuals, with one-third below 25 years of age, of whom half reported a severe level of impairment. A high prevalence of self-reported mental issues among adolescents and young adults has been reported in Norway [60], and there has been a significant increase in the number of young

individuals receiving disability benefits due to mental disorders in recent years [61]. The high proportion of younger adults in the current sample may thus reflect an improved effort to reach this group of patients to improve their level of functioning, which is important to prevent long-term sick leave and disability pension caused by mental disorders [62].

## Limitations

This study has several limitations that need to be acknowledged. First, the cross-sectional design prevents studying trajectories of functional impairment and sick leave status over time. Furthermore, the applicability of the findings is restricted to general outpatient clinics that offer routine care and does not extend to specialized units. Future research should employ longitudinal designs to explore whether WSAS is sensitive to changes that may occur during the treatment of mental disorders. The reported diagnoses were based on clinician assessment as part of routine treatment, however, to what extent this was based on the use of structured clinical interviews was not monitored. Additionally, the diagnosis-specific findings are somewhat uncertain due to a lack of inter-rater reliability. Employing standardized and controlled diagnostic procedures would enhance the accuracy and reliability of diagnoses. However, the study reports on naturalistic treatment settings based on standard procedures in outpatient clinics and provides clinically relevant information with high ecological validity. Sick leave status was self-reported, and although self-reported work status has demonstrated consistency with objective data registries [63] future studies should utilize register-based data. Another limitation of this study is that it did not include measures of symptom severity. Research from the same clinic as subsample 2 has shown that symptom severity is associated with health-related quality of life and that patients with personality disorders reported the lowest quality of life followed by trauma-related disorders, depression, and anxiety, while patients with ADHD reported the best quality of life [64]. Physical and psychological health are closely connected, and the study did not clarify the potentially differential impact of physical health illnesses compared to mental health diagnoses.

## Conclusion

This study examined the validity and reliability of WSAS among outpatients in routine mental health care. It supports WSAS as an important measure for assessing functional impairment in clinical settings, supplementing traditional symptom-based assessments in heterogeneous patient samples. Our findings reveal higher functional impairment among patients on sick leave compared to those in work, with variations across mental disorders. Depression was associated with the highest level of impairment, while patients with ADHD reported the lowest. Women reported more impairments in private leisure activities, while men reported challenges in maintaining close relationships. The results highlight the need to evaluate functional impairment as part of routine assessment in mental health care to tailor interventions and enhance routine mental health services. Our study advocates for future research using register data to broaden understanding and emphasizes the importance of evaluating treatment effectiveness based on both symptom alleviation and functional outcomes.

## Supporting information

**S1 Table. Factor loadings for WSAS when combining the two subsamples (50% $n_1$ and 50% $n_2$).**
(DOCX)

**S2 Table. CFA results for WSAS based on subsample 4 (50% $n_1$ and 50% $n_2$).**
(DOCX)

## Acknowledgments

We thank all the patients and clinicians at Nidaros DPS and Trondheim Kommune for participating in the current study.

## Author Contributions

**Conceptualization:** Jakob Lundqvist, Audun Havnen, Odin Hjemdal, Stian Solem.

**Data curation:** Jakob Lundqvist, Martin Schevik Lindberg, Martin Brattmyr.

**Formal analysis:** Jakob Lundqvist.

**Methodology:** Jakob Lundqvist.

**Supervision:** Audun Havnen, Odin Hjemdal, Stian Solem.

**Visualization:** Jakob Lundqvist.

**Writing – original draft:** Jakob Lundqvist, Martin Schevik Lindberg, Martin Brattmyr.

**Writing – review & editing:** Jakob Lundqvist, Martin Schevik Lindberg, Martin Brattmyr, Audun Havnen, Odin Hjemdal, Stian Solem.

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
