## [Decision Letter · Decision Letter 0]

10 Jan 2024

PONE-D-23-27409The Work and Social Adjustment Scale (WSAS): An investigation of reliability, validity, and associations with clinical characteristics in psychiatric outpatientsPLOS ONE

Dear Dr. Lundqvist,

Thank you for submitting your manuscript to PLOS ONE. After careful consideration, we feel that it has merit but does not fully meet PLOS ONE’s publication criteria as it currently stands. Therefore, we invite you to submit a revised version of the manuscript that addresses the points raised during the review process.

We look forward to receiving your revised manuscript.

Kind regards,

Ilias Mahmud, Ph.D.

Academic Editor

PLOS ONE

Journal Requirements: 

"This work was funded by the Norwegian University of Science and Technology and the Norwegian Labour and Welfare Administration (NAV) www.nav.no. The funding resulted in a PhD candidate position for JL. The funders had no role in study design, data collection and analysis, decision to publish, or preparation of the manuscript. "

We note that one or more of the authors is affiliated with the funding organization, indicating the funder may have had some role in the design, data collection, analysis or preparation of your manuscript for publication; in other words, the funder played an indirect role through the participation of the co-authors. If the funding organization did not play a role in the study design, data collection and analysis, decision to publish, or preparation of the manuscript and only provided financial support in the form of authors' salaries and/or research materials, please do the following:

a. Review your statements relating to the author contributions, and ensure you have specifically and accurately indicated the role(s) that these authors had in your study. These amendments should be made in the online form.

b. Confirm in your cover letter that you agree with the following statement, and we will change the online submission form on your behalf: 

“The funder provided support in the form of salaries for authors [insert relevant initials], but did not have any additional role in the study design, data collection and analysis, decision to publish, or preparation of the manuscript. The specific roles of these authors are articulated in the ‘author contributions’ section.

5. We note that you have indicated that there are restrictions to data sharing for this study. PLOS only allows data to be available upon request if there are legal or ethical restrictions on sharing data publicly. For more information on unacceptable data access restrictions, please see http://journals.plos.org/plosone/s/data-availability#loc-unacceptable-data-access-restrictions. 

Additional Editor Comments:

Please carefully follow the journal instructions to make sure that your manuscript adheres to the instructions.

Reviewers' comments:

Reviewer's Responses to Questions

**Comments to the Author**

1. Is the manuscript technically sound, and do the data support the conclusions?

Reviewer #1: Yes

Reviewer #2: Yes

2. Has the statistical analysis been performed appropriately and rigorously? 

Reviewer #1: Yes

Reviewer #2: No

3. Have the authors made all data underlying the findings in their manuscript fully available?

Reviewer #1: No

Reviewer #2: No

4. Is the manuscript presented in an intelligible fashion and written in standard English?

Reviewer #1: Yes

Reviewer #2: No

5. Review Comments to the Author

Reviewer #1: Overall impression. The study brings to attention important data that interest many spheres of psychology and psychotherapy. Although the study shows differences in the WSAS between different subgroups, nevertheless, for scale validation, it is important to also measure the invariance between the groups (the size of at least two groups would allow this).

- Is it possible to reproduce the web-based portal name?

- Participants and procedure: A table presenting the characteristics of the participants would be welcome.

Reviewer #2: The topic is fascinating, but a thorough restructuring of the content is necessary. I kindly request the esteemed author to revise the entire article with a deeper and more coherent perspective, taking into consideration the suggested changes. Afterward, please resubmit it for further review. I believe this feedback will be valuable in improving the quality of your work.

Best regards.

Abstract:

1. The objective could be more specific about how the study contributes to existing knowledge or addresses a gap. For example, is this the first time the WSAS is being validated in a Norwegian setting or in these specific mental health conditions?

2. results could be enriched with more specific data or statistics. For instance, mentioning the value of composite reliability could add more credibility to your claim of high internal consistency. Also, quantifying the effect sizes when discussing the largest impairments could make your findings more impactful.

3. Suggest areas where further research could build upon your findings. This shows a forward-thinking approach and situates your study within the broader research context.

introduction:

1. it could be more concise. Consider summarizing some of the background information to keep the focus on the main objectives of your study.

2. the specific problem your study addresses could be stated more clearly. Emphasize the gap in research or practice your study aims to fill.

3. Clearly state the primary objectives and hypotheses at the end of the introduction. This will provide a clear transition from the background to what your study specifically aims to achieve.

4. Highlight the relevance of your study to current clinical practices or policy implications in Norway's mental health care. This will underscore the importance of your research.

5. Provide a brief rationale for choosing your methodology, particularly why WSAS was selected for this study and its relevance in the Norwegian healthcare context.

6. Integrate studies more seamlessly into the narrative can enhance readability. Instead of listing studies, integrate them into a cohesive argument that builds towards your study's rationale.

7. Emphasize how your study adds to the existing literature. If your research addresses a limitation or gap not previously covered, make this clear to the reader.

Method:

• Clarify why specific patient groups were excluded (e.g., those with OCD, schizophrenia, etc.). Explain how these exclusions might impact the generalizability of your findings.

• Mention any ethical approvals obtained for the study, especially since it involves human subjects.

• While you have explained WSAS well, briefly discuss its historical validity and reliability to establish its credibility as a measurement tool.

• Provide a rationale for choosing EFA and CFA, and explain why these methods are appropriate for your study.

• Elaborate on the strategy for handling missing data beyond just removing non-responders. This is important for the validity of your results.

• While you have listed various fit indices, explain what each one indicates and why it is relevant to your analysis.

• When discussing group differences, specify how you controlled for potential confounders (if applicable).

Discussion

1. RMSEA value indicated a poor model fit, yet the SRMR value suggested a close-fitting model. Discuss why there is this discrepancy and what it implies for the validity of your model

2. Compare reliability coefficients with those reported in previous studies, if available.

3. Begin by explicitly linking your discussion to the study's objectives and key findings. This establishes a clear framework for your analysis.

4. Elaborate on the implications of the one-factor model and good internal consistency for using WSAS in routine mental health care. Discuss how these findings contribute to the existing knowledge.

5. Explore the reasons behind the observed differences in functional impairment based on sick leave status, psychiatric diagnoses, and sex. Discuss how these findings align or contrast with existing research.

6. Compare your results with previous studies more explicitly. Where your findings agree or differ, discuss possible reasons for these similarities or discrepancies.

7. Discuss the practical implications of your findings for mental health practitioners. How can these results inform clinical practice, especially in the context of assessing and addressing functional impairment?

8. While you've identified some limitations, consider discussing how these might affect the interpretation of your results and the generalizability of your findings.

9. Suggest specific areas for future research based on your findings and limitations. This could include longitudinal studies, the use of structured clinical interviews, or exploring functional impairment in other patient groups.

10. If relevant, discuss how cultural or societal factors might influence the reported differences in functional impairment, especially in the context of gender and mental health.

11. Offer specific recommendations for mental health policy or practice based on your findings. How might routine assessments be improved?

12. Conclude the discussion by summarizing the main points, emphasizing the significance of your study, and restating its contribution to the field.

6. PLOS authors have the option to publish the peer review history of their article (what does this mean?). If published, this will include your full peer review and any attached files.

Reviewer #1: No

Reviewer #2: **Yes: **Roghieh Nooripour

---

## [Author Response · Author response to Decision Letter 0]

7 Feb 2024

Dear Editor, Dr. Ilias Mahmud and reviewers,

Thank you for your valuable feedback on our manuscript. Your valuable feedback has been carefully considered, and we have made the necessary revisions to address the comments provided in this point-by-point rebuttal letter. We appreciate your time and consideration of our revised manuscript and look forward to your reply. 

Jakob Lundqvist, jakob.lundqvist@ntnu.no

NTNU, Department of Psychology

---

## [Decision Letter · Decision Letter 1]

20 Aug 2024

PONE-D-23-27409R1The Work and Social Adjustment Scale (WSAS): An investigation of reliability, validity, and associations with clinical characteristics in psychiatric outpatientsPLOS ONE

Dear Dr. Lundqvist,

Thank you for submitting your manuscript to PLOS ONE. After careful consideration, we feel that it has merit but does not fully meet PLOS ONE’s publication criteria as it currently stands. Therefore, we invite you to submit a revised version of the manuscript that addresses the points raised during the review process.

Please address the comments from the reviewers, particularly those from Reviewer 3, and resubmit your manuscript following the guidelines provided below.

We look forward to receiving your revised manuscript.

Kind regards,

Ilias Mahmud, Ph.D.

Academic Editor

PLOS ONE

Journal Requirements:

Reviewers' comments:

Reviewer's Responses to Questions

**Comments to the Author**

1. If the authors have adequately addressed your comments raised in a previous round of review and you feel that this manuscript is now acceptable for publication, you may indicate that here to bypass the “Comments to the Author” section, enter your conflict of interest statement in the “Confidential to Editor” section, and submit your "Accept" recommendation.

Reviewer #1: All comments have been addressed

Reviewer #2: All comments have been addressed

Reviewer #3: (No Response)

2. Is the manuscript technically sound, and do the data support the conclusions?

Reviewer #1: Yes

Reviewer #2: Yes

Reviewer #3: Yes

3. Has the statistical analysis been performed appropriately and rigorously? 

Reviewer #1: Yes

Reviewer #2: Yes

Reviewer #3: Yes

4. Have the authors made all data underlying the findings in their manuscript fully available?

Reviewer #1: Yes

Reviewer #2: Yes

Reviewer #3: Yes

5. Is the manuscript presented in an intelligible fashion and written in standard English?

Reviewer #1: Yes

Reviewer #2: Yes

Reviewer #3: Yes

6. Review Comments to the Author

Reviewer #1: I consider that the manuscript has been perfected and that the authors have addressed all the requirements.

Reviewer #2: The revisions to your paper have been diligently implemented, resulting in a fully corrected and now acceptable manuscript. We extend our gratitude for your dedicated efforts and collaborative approach in elevating the overall quality of your article.

We wish you continued success in your forthcoming research and writing endeavors.

Reviewer #3: Ref.: PONE-D-23-27409R1

Title: The Work and Social Adjustment Scale (WSAS): An investigation of reliability, validity,

and associations with clinical characteristics in psychiatric

PLOS ONE

The present study examines the validity and reliability of the WSAS in Norway. A strength of the study is the use of large sample utilized for the study. The authors are commended for examining the psychometric properties of the scale in different cultures, languages, and settings in which the scale was originally developed as this important issue is routinely neglected. Recommendations to improve the investigation are reviewed below.

1. Abstract. It is unclear why the authors chose to do the EFA on 1/3 and CFA on 2/3 or sample? Why not use ½ of the sample for EFA and the other ½ for the CFA as is the common practice? The sample size is large enough that I am not concerned about the results being negatively impacted due to think, but this is more of a reflection to consider for future investigations.

2. Pg 4. It would be helpful to flesh out the paragraph on differences in impairment across disorders. Does the literature describe reasons for the greater impairment in some groups (e.g., personality disorder) than others? How does comorbidity or disorder severity impact these patterns?

3. Pg 9, setting section. Given the differences in the two sample (sample 1 – mild symptoms, sample two – moderate to severe symptoms) is seems that combining the samples and then conducting the EFA on ½ of the full sample and the other ½ of the combined sample would be a better approach for conducting the EFA and CFA to allow for more variation in symptom level across that EFA and CFA rather than truncating mild symptoms to EFA and moderate to severe in the CFA. Some rationale for the approach the authors used is needed.

4. Methods / analyses – Do the authors know the reason for why people were on sick leave? Was the leave related to mental health problems? What about physical health issues? Are the authors able to report on level of physical health issues in the sick leave group. If not, this should be listed as a key confound variable as differences in physical health problems in the sick leave group may be driving the results more than the specific mental health diagnoses.

5. Results. It looks like comorbidity was addressed in the analyses, but where there differences in the overall severity of patient groups?

6. Pg 15, 1st paragraph. I’m not convinced this study is using the most “heterogenous samples” for EFA and CFA compared to other studies. As mentioned above, the EFA was conducted on sample with mild symptoms and CFA on moderate to severe symptoms. By analyzing these separately, the individual samples are less heterogenous than if the authors would have combined the two samples and conducted EFA on one ½ of the sample and CFA on the other ½. This would have allowed for the modeling of symptom range from mild to severe in both the EFA and CFA, rather than mild in EFA and mod/severe in the CFA.

7. General discussion comment. Please add issue related to the impact of physical health illnesses impacting the sick leave group. We cannot assume that it is only the mental health diagnoses that are driving the results in the sick leave group.

7. PLOS authors have the option to publish the peer review history of their article (what does this mean?). If published, this will include your full peer review and any attached files.

Reviewer #1: No

Reviewer #2: **Yes: **Roghieh Nooripour

Reviewer #3: No

---

## [Author Response · Author response to Decision Letter 1]

3 Sep 2024

Thank you for your thorough and insightful comments. As outlined in the point-by-point rebuttal letter, we have carefully addressed the concerns raised by the reviewers. We greatly appreciate your time and consideration of our revised manuscript and look forward to your feedback.

---

## [Decision Letter · Decision Letter 2]

18 Sep 2024

The Work and Social Adjustment Scale (WSAS): An investigation of reliability, validity, and associations with clinical characteristics in psychiatric outpatients

PONE-D-23-27409R2

Dear Dr. Lundqvist,

We’re pleased to inform you that your manuscript has been judged scientifically suitable for publication and will be formally accepted for publication once it meets all outstanding technical requirements.

Kind regards,

Ilias Mahmud, Ph.D.

Academic Editor

PLOS ONE

Additional Editor Comments (optional):

Reviewers' comments:

Reviewer's Responses to Questions

**Comments to the Author**

1. If the authors have adequately addressed your comments raised in a previous round of review and you feel that this manuscript is now acceptable for publication, you may indicate that here to bypass the “Comments to the Author” section, enter your conflict of interest statement in the “Confidential to Editor” section, and submit your "Accept" recommendation.

Reviewer #3: All comments have been addressed

2. Is the manuscript technically sound, and do the data support the conclusions?

Reviewer #3: Yes

3. Has the statistical analysis been performed appropriately and rigorously? 

Reviewer #3: Yes

4. Have the authors made all data underlying the findings in their manuscript fully available?

Reviewer #3: Yes

5. Is the manuscript presented in an intelligible fashion and written in standard English?

Reviewer #3: Yes

6. Review Comments to the Author

Reviewer #3: Thank you for addressing my comments. The comments have been addressed and I do not have any additional comments.

7. PLOS authors have the option to publish the peer review history of their article (what does this mean?). If published, this will include your full peer review and any attached files.

Reviewer #3: No

---

## [Editor Report · Acceptance letter]

2 Oct 2024

PONE-D-23-27409R2 

PLOS ONE

Dear Dr. Lundqvist, 

I'm pleased to inform you that your manuscript has been deemed suitable for publication in PLOS ONE. Congratulations! Your manuscript is now being handed over to our production team.

Kind regards, 

on behalf of

Dr. Ilias Mahmud 

Academic Editor

PLOS ONE